## Trauma-informed approaches to primary and community mental health care: protocol for a mixed-methods systematic review

Shoba Dawson ![ORCID],[1] Angel Bierce,[1] Gene Feder,[1] John Macleod,[1,2,3] Katrina M Turner ![ORCID],[1,2,3] Stan Zammit,[3,4] Natalia V Lewis ![ORCID] [1,2,3]

For numbered affiliations see end of article.

**Correspondence to**
Dr Shoba Dawson;
shoba.dawson@bristol.ac.uk

## ABSTRACT

**Introduction** Exposure to different types of psychological trauma may lead to a range of adverse effects on trauma survivors, including poor mental and physical health, economic, social and cognitive functioning outcomes. Trauma-informed (TI) approaches to care are defined as a service system grounded in and directed by an understanding of how trauma affects the survivors' neurological, biological, physiological and social development. TI service system involves training of all staff, service improvements and sometimes screening for trauma experiences. The UK started incorporating TI approaches into the National Health Service. While policies recommend it, the evidence base for TI approaches to healthcare is not well established. We aim to conduct a systematic review to synthesise evidence on TI approaches in primary and community mental healthcare globally.

**Methods and analysis** We will undertake a systematic search for primary studies in Medline, Embase, PsycINFO, Cumulative Index to Nursing and Allied Health Literature, Cochrane library, websites of organisations involved in the development and implementation of TI approaches in healthcare, and databases of thesis and dissertation. Included studies will be in English published between 1990 and February 2020. Two reviewers will independently perform study selection with data extraction and quality appraisal undertaken by one reviewer and checked for accuracy by a second reviewer. A results-based convergent synthesis will be conducted where quantitative (narratively) and qualitative (thematically) evidence will be analysed separately and then integrated using another method of synthesis. We set up a trauma survivor group and a professional group to consult throughout this review.

**Ethics and dissemination** There is no requirement for ethical approval for this systematic review as no empirical data will be collected. The findings will be disseminated through a peer-reviewed publication, scientific and practitioner conferences, and policy briefings targeted at local and national policy makers.

**PROSPERO registration number** CRD42020164752.

## Strengths and limitations of this study

► This is the first mixed-methods systematic review of trauma-informed (TI) approaches in primary and community mental healthcare.
► The review will include peer-reviewed and grey literature, providing a global view of TI approaches for informing healthcare services and future research in the UK.
► The involvement of people with lived experiences of trauma and healthcare professionals helps to produce evidence that is relevant to providers and recipients of healthcare and is therefore, more likely to be translated into practice.
► One limitation of the review is that it will exclude studies without an abstract in English which could lead to missing relevant studies.
► The review will not include policy documents. This might be an area for future research.

trauma-informed (TI) care, defines individual trauma as an 'event, series of events, or set of circumstances that is experienced by an individual as physically or emotionally harmful or life-threatening and that have lasting adverse effects on the individual's functioning and mental, physical, social, emotional or spiritual well-being'.[1] According to the WHO World Mental Health Survey, 70% of respondents experienced lifetime traumas, with exposure averaging 3.2 traumatic events per person. The most frequently reported traumas were those that occurred to loved ones/witnessed (36%), those involving accidents (34%), unexpected death of loved ones (31%), physical violence (23%), intimate partner sexual violence (14%) and war-related traumas (13%).[2] In the English household survey, 47% of adults reported at least one adverse childhood experience (ACE). Prevalence of childhood sexual, physical and verbal abuse was 6%, 15% and 18%, respectively.[3] According to the Crime Survey

## INTRODUCTION

The Substance Abuse and Mental Health Services Administration (SAMHSA), which is the leading institution in the field of

for England and Wales, 8% of women and 4% of men experienced domestic and sexual violence and abuse in the last year; lifetime prevalence was 29% and 13%, respectively.[4]

Experiencing trauma can have a wide range of adverse impacts on the victims, including poorer mental health, physical health, economic and social outcomes throughout the life span.[5] This means that a large proportion of people with health problems who access primary healthcare and community mental healthcare have experienced trauma in their lifetime. Primary healthcare and community mental healthcare is the first point of contact with a health system for an individual. It plays a vital role in making healthcare universally accessible.[6] The WHO has adopted primary care as the preferred method for providing comprehensive, equitable, affordable and universal healthcare services for individuals and communities.[6] The WHO has made a substantial investment to ensure that mental health services are integrated into primary care in the last decade.[7] The rationale for the integration of mental health services into primary care includes: reduced stigma, improved access to care, reduced chronicity and improved social integration, better health outcomes for people treated in primary care and improved human resource capacity for mental health.[7 8] Both primary healthcare and community mental healthcare provide more accessible outpatient services. Both deal with patients who have co-occurring conditions and multiple health and social needs. Both integrate patient care across medical specialities and varied service providers.

Several studies found strong evidence on the association between lifetime traumas and increased utilisation of primary healthcare. The household survey in England and Wales found that adults with four ACEs were twice as likely to visit a general practitioner six times or more in the last 12 months (OR 2.3, 95% CI 1.8 to 2.9) compared with adults with no ACEs.[9] The Australian Longitudinal Study of Women's Health found that women with lifetime violence experiences had almost twice the odds of higher general practice service use (Adjusted Odds Ratio (AOR) 1.82, 95% CI 1.37 to 2.40), compared with women without any violence experiences.[10] A majority of patients in community mental health and substance abuse services experienced repeated trauma throughout their life span.[11] Therefore, primary healthcare services must be designed in a way that will support the recovery of survivors. However, they often have the opposite effect and can trigger memories about traumatic experiences through invasive procedures and coercive practices (eg, the removal of choice regarding treatment or judgmental attitudes following a disclosure of abuse; lack of available and acceptable services).[12] Re-activation of traumatic experiences within health services can affect both service users and staff, with the latter experiencing vicarious trauma.[13]

The field of TI approach to care (synonyms *TI care, TI practice, TI model of care, TI service system*) is relatively new.

The concept of TI approach was developed in the USA,[14] where it is widely used across all sectors.[1] Various organisations, expert panels and researchers proposed varied definitions of TI care. One of the consensus-based definitions describes TI care as 'a strengths-based service delivery approach that is grounded in an understanding of and responsiveness to the impact of trauma, that emphasises physical, psychological and emotional safety for both providers and survivors, and that creates opportunities for survivors to rebuild a sense of control and empowerment'.[15] Through consultations with the expert panel, SAMHSA developed a flexible framework comprising foundation assumptions, principles and implementation domains for a TI care[1] that can be adapted to any service system including primary healthcare.[16] SAMHSA's framework of TI approach is based on the four key assumptions of:

1. Recognition: all people in an organisation recognise how patients' and staff's experiences of trauma might affect the way they think, feel and behave.
2. Realisation: all staff in the organisation accept how trauma can affect people, and patient's behaviour is understood in the context of coping with their experiences.
3. Response: the organisation acts to effectively integrate knowledge about prevalence and impact of trauma into policies, procedures, and practices.
4. Resist the re-activation of traumatic memories: steps are taken to prevent further traumatising both service users and staff through a focus on the recovery of survivors, as well as the well-being of staff.[1]

SAMHSA's TI approach framework includes six key principles applicable to varied settings: (i) safety, (ii) trustworthiness and transparency, (iii) peer support, (iv) collaboration and mutuality, (v) empowerment, choice and choice and (vi) cultural/historical/gender issues.

A TI approach is distinct from trauma-specific interventions (eg, trauma-focused cognitive behavioural therapy) or trauma services (eg, Traumatic Stress Service) that treat trauma symptoms. A TI approach can include trauma-specific interventions, although the essential component is the application of the above assumptions and principles in the organisational/system levels.[1]

Most extant evidence for TI approaches comes from Northern America.[17–19] TI care has only recently been included in the UK National Health Service (NHS) long-term plan[20] and the NHS mental health implementation plan.[21] The Scottish Government[22] and the Safeguarding Board for Northern Ireland[23] endorsed TI approaches across healthcare, social care, education and justice sectors. Public Health Wales published reports recommending TI approaches across public services.[24] TI approaches were endorsed in local governments policies across England.[25] However, the evidence base for TI approaches to healthcare is not well established. A recent scoping review identified only a few examples of TI care implemented in Scotland[22] and England[26 27] and

recommended developing the evidence base to demonstrate the value of TI approaches in the UK healthcare context.[28]

The aim of this systematic review is to synthesise evidence on TI approaches in primary care and community mental healthcare globally, which will help inform the development of a UK specific model of TI in these settings. The synthesis will address the following research questions:

1. What models of TI care have been used in primary care and community mental health services?
2. What are the formal theories and empirically supported theories of change underpinning these models and their evaluations?
3. What evidence is available for the acceptability, effectiveness and cost-effectiveness of TI approaches to primary care and community mental healthcare?

## METHODS AND ANALYSIS
This protocol follows the Preferred Reporting Items for Systematic Review and Meta-Analysis Protocols checklist (online supplemental appendix 1).[29] The systematic review will be conducted and reported following the Preferred Reporting Items for Systematic Reviews and Meta-Analyses (PRISMA) guidelines.[30]

### Eligibility criteria
We will select studies according to the following criteria.

### Population
Patients in primary and community mental healthcare (globally) aged 18 and over. Professionals who plan, commission and deliver primary or community mental healthcare.

### Intervention
We will use the SAMHSA definitions of TI approach[1] and include evaluations of any model of TI care in primary or community mental health organisations. The SAMHSA framework of TI care covers ten implementation domains of organisational change in (i) governance and leadership, (ii) written policies and protocols, (iii) physical environment, (iv) training and workforce development, (v) engagement and involvement of service users, (vi) cross-sector collaboration, (vii) progress monitoring and quality assurance, (viii) financing, (ix) evaluation, (x) screening, assessment and treatment for trauma.

We will exclude studies of trauma-specific interventions that treat trauma symptoms.

### Comparator
Primary and community mental healthcare not using TI approaches.

### Outcomes
We identified outcomes from the prior literature on TI approaches to health care[17 31–34] and consultations with two study advisory groups of trauma survivors and professionals who plan, commission and deliver primary and community mental healthcare. To map the outcomes from all these sources on the draft logic model of a TI care, we adapted a published measurement model for TI primary care (figure 1).[34] Throughout review process, we will refine this draft logic model in discussions within the team and consultations with the study advisory groups.

To be included, a study must report a measure from at least one of the above outcome categories. The TI approach is an organisation level intervention. Therefore, we define outcomes at the organisation level as main outcomes and outcomes at the individual level as additional outcomes. We will also look at formal theories and empirically supported theories of change underpinning included TI approaches and their evaluations. We will pay special attention to adverse effects of TI approaches on patient health, healthcare providers (eg, vicarious trauma), service utilisation and quality of care.

### Types of studies
Primary studies of any design that evaluated the acceptability and/or effectiveness and/or cost-effectiveness of TI approaches in primary and/or community mental healthcare will be included. We will include studies with mixed samples only if outcomes for the primary healthcare and/or community mental healthcare subsample are reported separately irrespective of the proportion of the subsample. Reference lists of systematic reviews that meet this criterion will be searched to identify relevant primary studies. Editorials, policy documents and books will be excluded.

### Setting
Any setting providing primary care, including primary care mental health services. WHO defines a primary healthcare centre as setting providing services that are usually the first point of contact with a healthcare professional.[35] Depending on the country, they can include any open access, community based first point of care service, for example, general practice clinics, community-based units, basic health units, family health strategy, primary care home visits, day-care centres, multicentre health clinics, one stop crisis centre, improving access to psychological therapies services.[8]

### Time frame
An early and influential paper[14] discussing TI approaches was published in 2001. However, 1990 was chosen as a starting point to capture any relevant and early discussions of TI care principles from a global perspective. We will limit the studies by date to ensure that the search identifies all relevant studies since this publication.

### Language
There will be no language restrictions, provided an English language abstract is available for initial screening. During full-text screening, if the included papers are not available in English, we will translate them with help from multilingual colleagues and Google Translate.

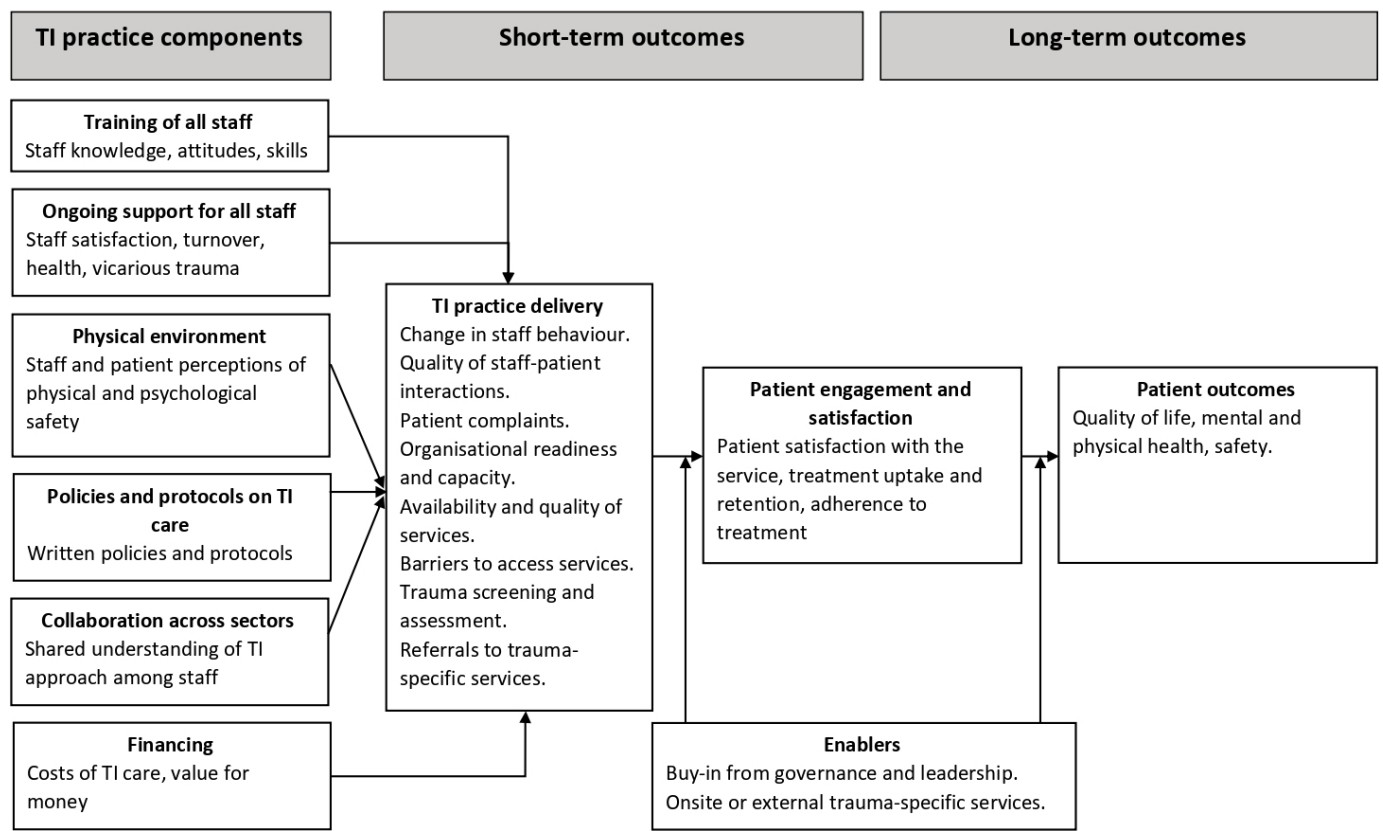

**Figure 1** Draft logic model of trauma-informed primary and community mental healthcare. TI, trauma-informed. Adapted from Germán.[34]

## Search strategy

SD and NVL will develop a comprehensive search strategy using a combination of MeSH and free-text terms, based on previous systematic reviews in the areas of TI care[32 36] and the expertise of the research team to identify relevant papers on TI approaches in primary and community mental healthcare for adults. SD will run several scoping exercises in different electronic databases to maximise the sensitivity and specificity of the developed search strategy (online supplemental appendix 2, example search strategy).

SD will search electronic bibliographic databases for potential primary studies from January 1990 to February 2020: Cochrane Library, MEDLINE, EMBASE, Cumulative Index to Nursing and Allied Health Literature (CINAHL EBSCO) and PsycINFO and update searches in the last 2 months of the study.

In addition, SD will search the PROSPERO database to identify any relevant systematic reviews in progress. She will also conduct a grey literature search to identify studies not indexed in the databases listed earlier. SD will search websites of organisations involved in development and implementation of TI care: UK national and local governments, King's Fund, SAMHSA, Violence, Abuse and Mental Health Network and Trello (Adverse Childhood Experiences Resource library). Theses and dissertations will be identified through ethos library and PROQUEST. SD, AB, NVL will carry out forward and backward referencing of included papers to supplement the database and grey literature searches to identify any further relevant articles.

SD and NVL will approach corresponding authors of included papers, study advisory groups and experts in the field of trauma and primary care for additional relevant articles.

## Screening of studies

References will be managed in Rayyan (https://rayyan.qcri.org/). SD will export search results from the different databases into the Rayyan database and remove duplicates. Study selection will be completed in two stages: first, titles and abstracts will be screened; next full-text will be screened to identify studies eligible for inclusion. Two members of the research team (SD and AB or NVL) will independently screen titles, abstracts and full-text. Any discrepancies between reviewers will be discussed with other team members.

We will list excluded full text studies in the table categorised by reasons for exclusion. We will collate multiple reports of the same study so that each study, rather than each report, is the unit of analysis in the review. We will also provide any information we can obtain from corresponding authors about ongoing studies. We will record the screening process in detail to complete a PRISMA flow diagram.[37]

## Data extraction

A data extraction form will be used to focus on the characteristics that are relevant to this review:

1. Methods: type of study (randomised trial, interrupted time series, controlled/uncontrolled before-after, cross-sectional, qualitative, mixed-method, service evaluation).
2. Study setting (country, key features of the healthcare system, healthcare setting).
3. Characteristics of the participants (age, sex, ethnicity, condition as described and identified by the authors of included studies).
4. Characteristics of the TI approach: components, comparison. We will map components of each TI model on the SAMHSA's 10-domains framework of organisational change.[1]
5. Outcomes: main and additional outcomes specified and collected, time points reported.
6. Theories underpinning the TI approach: formal theories and/or empirically developed theories of change explaining how the intervention works. We will also extract data on theories underpinning included evaluations.

We will seek input from the study advisory groups of trauma survivors and professionals on any other relevant data that should be extracted. SD will pilot the adapted extraction form on publications of a quantitative, qualitative and mixed-method study, and then refine it. To minimise bias and errors, one reviewer (SD) will extract the data and a second reviewer (AB or NVL) will check the extraction in detail. Any disagreements between reviewers will be resolved through discussions and, if required, with other team members. We will ask corresponding authors of included studies to check reconciled data extraction forms and provide missing information and clarifications.

## Quality appraisal

We will use the Mixed Methods Appraisal Tool (MMAT) tool to appraise the studies.[38] Quality appraisal will be carried out as part of data extraction. SD will complete MMAT checklists for each study. Second reviewer (AB or NVL) will check completed checklists in detail. Any disagreement between the reviewers will be resolved through discussions and if required by other members of the team.

## Data synthesis
### Synthesis of quantitative data

Based on feasibility searches and background reading, we expect that included quantitative studies will report outcomes that vary substantially by the way they were defined and measured. For this reason, we anticipate that a quantitative synthesis of the results from the quantitative studies will not be appropriate. We will, therefore, employ narrative synthesis to summarise findings from quantitative studies. This will involve the use of descriptive text and tables to summarise data to allow readers to consider findings in the light of differences in study designs. We will describe all TI models in a table based on the TiDIER template.[39] For each model of TI care, we will describe the range of effects found in the studies and if possible, the theory of change through which the TI models were intended to affect specific outcomes.

### Synthesis of qualitative data

For qualitative studies, we plan to use the thematic synthesis method.[40]

### Synthesis of quantitative and qualitative findings

We will use a results-based convergent synthesis design,[41] following the Sandelowski's segregated method.[42] First, we will analyse and synthesise the quantitative (narrative synthesis) evidence and the qualitative (thematic synthesis) separately as described earlier. Next, we will then integrate the synthesis products (results of both syntheses) using another method of synthesis (eg, tables, matrices or reanalysing evidence as a result of both syntheses), which allows for comparing and/or juxtaposing the findings from the quantitative and qualitative evidence.[41] At this stage, we will map all the evidence on the refined logic model of TI care (figure 1) and finalise it through discussions within the team and consultations with the survivor and professional advisory groups.[43]

## DISCUSSION

An effective response to (often) hidden trauma of patients in general practice and primary care mental health services is long overdue. This is a protocol for a systematic review on TI approaches in primary care and community mental healthcare that addresses the gap in evidence on the acceptability, effectiveness and cost-effectiveness of TI healthcare. The study includes both peer-reviewed and grey literature and offers a global view of TI approaches. Our findings will inform the development of an evidence-based UK-specific model of TI primary care and community mental healthcare. Although the output of this review will form the basis for further research, the findings will be relevant to current policy and practice, even before we have developed and tested the UK-specific TI model. UK policymakers can use this new evidence when developing/amending health policies on TI care. Involvement of two advisory groups of trauma survivors and providers of healthcare throughout all stages of this review helps to produce evidence that is relevant to end-users and is likely to be translated into policy and practice.

A limitation of this review is the use of search terms based on the current TI terminology, which was introduced in early 2000 (online supplemental appendix 2). We might miss the earlier studies which evaluated healthcare services with TI approaches that were not labelled as such. We addressed this limitation by designing a search strategy with input from our advisory groups. We included a term for psychologically informed environments from the pre-TI era and also undertook searches from 1990. Another limitation of the review is that it will exclude studies without an abstract in the English language, which could lead to missing relevant studies. Policy documents will be excluded from this review and is an area for a future policy review.

This study was conceived and designed at the pre-COVID-19 pandemic era and will be delivered and disseminated throughout and after the pandemic. The WHO[44] and statutory and third sector organisations[45 46] have already reported that stress, social isolation measures, quarantines at home and so on resulted in the increase of all forms of family violence. Consultation with our advisory groups on the impact of the pandemic has supported these findings. Our lay and professional contributors talked about the rise in traumatic experiences among patients and healthcare professionals, worsening mental health, increased demand for health and social services and transition of services from face-to-face delivery to phone or online delivery mode. This changing environment makes our systematic review of TI approaches to healthcare timely in providing evidence for effective and acceptable primary and community mental healthcare in the post-pandemic era.

## Patient and public involvement

We have set up two advisory groups to work with researchers throughout the study: a trauma survivor group and a professional group. The trauma survivor advisory group consists of eight people with diverse lived experiences of trauma who had been recipients of care in the NHS and other care systems. The professional advisory group consists of eight professionals (eg, planning and development manager from the local authority, clinical psychologist, clinical lead specialist) from England and Wales who are involved in planning, funding, commissioning or delivering primary healthcare or community mental health services.

We organised separate meetings with the survivor and professional advisers where they were introduced to the study and systematic review process. Both groups took part in brainstorming exercises on formulating research questions and listing outcomes for the systematic review that are meaningful to patients, practitioners, service managers and commissioners. Based on their feedback, SD and NL mapped the outcomes to those identified in the existing literature independently, and any discrepancies were resolved through discussions. The professional advisory group developed a list of UK primary and community mental health services. The group highlighted inconsistent terminology used in the UK (ie, ACEs, TI approaches/care/practice, psychologically informed environments) and diverse approaches to developing policies on TI care across the UK (ie, top-down in Scotland, Northern Ireland and Wales vs bottom-up in England). They highlighted the need for synthesised research evidence about such approaches that are relevant to the UK and for using this evidence to inform health policy. We will meet with the survivor and professional advisory groups biannually to consult on data extraction, interpretation, and dissemination of study findings.

## ETHICS AND DISSEMINATION

No ethical approval is required for this systematic review, as this does not involve the collection of primary data. Findings of this review will be disseminated through publication of a peer-reviewed paper, papers presented at conferences for academic and practitioner audiences, through local clinical commissioning groups and policy briefings targeted at local and national policymakers. Our survivor and professional advisers will be consulted for sources through which we should disseminate the findings. We will also work with survivor advisers to develop a lay summary of the study findings which will be disseminated through the Centre for Academic Primary Care PPI contributors. Finally, we will produce a policy brief of the study findings for dissemination among professional stakeholders involved in planning, funding, commissioning and delivery of primary and community mental healthcare.

**Author affiliations**
[1]Centre for Academic Primary Care, Population Health Sciences, University of Bristol, Bristol, UK
[2]Centre for Academic Primary Care, Population Health Sciences, National Institute for Health Research Applied Research Collaboration West (ARC West), Bristol, UK
[3]Centre for Academic Primary Care, Population Health Sciences, National Institute for Health Research (NIHR) Bristol Biomedical Research Centre, University Hospitals Bristol NHS Foundation Trust, University of Bristol, Bristol, UK
[4]Division of Psychological Medicine and Clinical Neurosciences, Cardiff University, Cardiff, UK

**Acknowledgements** We would like to thank Mike Bell (BRC PPI coordinator) for supporting us with the set up and coordination of study advisory groups and our survivor and professional advisory group for their contribution in identifying outcomes for this systematic review.This research was supported by the National Institute for Health Research (NIHR) (17/63/125) using UK aid from the UK Government to support global health research. The views expressed in this publication are those of the authors and not necessarily those of the NIHR or the UK government.

**Contributors** NVL conceived the idea and secured funding. NVL and SD designed the study. SD drafted the study protocol. SD, AB and NVL set up and coordinate study advisory groups. SD and NVL produced first draft of the manuscript. All coauthors (SD, AB, GF, JM, KMT, SZ, NVL) contributed to the refinement of the manuscript and approved the final version for submission.

**Funding** This study is funded by the NIHR Biomedical Research Centre at University Hospitals Bristol NHS Foundation Trust and the University of Bristol. Grant reference number R100514-105. The views expressed are those of the authors and not necessarily those of the NIHR or the Department of Health and Social Care.

**Competing interests** None declared.

**Patient consent for publication** Not required.

**Provenance and peer review** Not commissioned; externally peer reviewed.

**ORCID iDs**
Shoba Dawson http://orcid.org/0000-0002-6700-6445
Katrina M Turner http://orcid.org/0000-0002-6375-2918
Natalia V Lewis http://orcid.org/0000-0002-4839-6548

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
