## [Reviewer comments · BMJ Open]

ARTICLE DETAILS

TITLE (PROVISIONAL)	Trauma-informed approaches to primary and community mental health care: protocol for a mixed-methods systematic review
AUTHORS	Dawson, Shoba; Bierce, Angel; Feder, Gene; Macleod, John; Turner, Katrina; Zammit, Stan; Lewis, Natalia

VERSION 1 – REVIEW

REVIEWER	JOAN MARIE HALIBURN UNIVERSITY OF SYDNEY SYDNEY, AUSTRALIA
REVIEW RETURNED	25-Oct-2020

GENERAL COMMENTS	This study promises to be an innovative contribution to the mental health literature. Trauma-informed-care are the catch words in current mental health services, but there is no certainty that they are truly trauma informed i.e. adhering to the principles of such an approach. It will be a step towards refining such principles, leading to a more global understanding and approach. The study proposes to use the definition of trauma as proposed by SAMHSA, the leading institution in the field of trauma-informed-care. Do you think their definition will suit other mental health services entirely? The authors have wisely chosen to obtain data from primary mental health services - they are usually the first point of contact, and the amount of data gathered will be significant to yield valid results, and the diversity of practitioner will enrich it. A systematic review of synthesized evidence in trauma-informed approaches to health care in primary and community mental health facilities globally, is necessary, in fact imperative, if we are to truly deliver timely assessment and treatment without re-traumatizing individuals who come for help, or forgetting workers who may run the risk of vicarious traumatization. This review is indeed timely and can be an innovative contribution to the mental health care literature and to mental health care in general. Trauma-informed care is an emergent paradigm, but the current organization of health and human services does not reflect this reality and is inadequate, thus a common finding is one of trauma occurring in a service itself (Jennings 2004). Results of a well-thought through study will help to make sure that services carry a detailed definition of what it is to be trauma-informed and what it is to deliver a service that adheres to this definition. The impact of trauma was recognized decades ago, but the roots of trauma-informed care are more recent, and we must be sure that it delivers this care based on well-informed principles. Trauma-informed care is being incorporated in policies, world-wide, but where is the evidence? This review seeks to provide us
---

	with the evidence. This study has sought input from those with lived experience of trauma, and professionals who work with those who have been traumatized. In doing so it has respected the knowledge that sufferers themselves have experienced – both trauma as well as care – thus being able to attest to the kind of care they have received, and whether they felt it was adequate or not – the flaws along with the positive aspects and suggestions for improvement. It also intends to seek the personal experiences of professionals who have delivered services. Professionals similarly can be gratified to know that their interests are also in the minds of employment systems. However, this study seeks to adopt the definition of trauma-informed care used by SAMHSA the leading institution in this field. In this definition, the words psychological, emotional and behavioural are missing (Dawson et. al 2020: p. 5, lines 16- 22), though this is made good in the Introduction (Dawson et.al 2020: p. 7 lines 9-18). They are the core issues in both assessment and treatment of patients in the mental health system. SAMHSA since 1984 has pursued the need for trauma-informed services and has since become a major vehicle for the advocacy of Trauma Informed Care in the USA. Trauma “has often occurred in the service context itself” (Jennings, 2004: p.6). Will the principles of trauma-informed care provided by drug and alcohol services fit with the principles of those facilities providing care of the rest of the mentally ill? The choice of primary health care and community health care as sources of study is a good one, as increased health care utilization is strongly connected with early childhood traumas. Therefore, the need to take particular precautions in assessment and treatment are vital, and so is the need to prevent vicarious re-traumatization. Trauma informed approaches to care shift the health care professional’s focus from diagnosis towards understanding trauma that lies at the root of symptoms and health conditions (page 9, lines 17-43). Awareness of a trauma history, or awareness of trauma in a person’s mental state examination needs to be borne in mind and addressed alongside the presenting condition - but being seen as “the root of symptoms and health conditions” deserves a more cautious approach. We still do not have the evidence that trauma is at the root of Schizophrenia and other Major Mental Disorders, even though the associations are high. Principles of Trauma-Informed Care must take priority in any trauma-informed service, and adherence to such principles can become apparent in the manner in which organizations teach, train and observe such principles, both towards their staff as well as in the delivery of services to their patients. Observations of principles such as safety, trust, choice, collaboration and empowerment are necessary if a service is to be truly trauma -Informed. This could do with some elaboration in the study. This review on the whole takes into consideration the essentials required in trauma-informed services which are widely advocated, but require sound evidence in its implementation. References:
--	--

	 1. Dawson, S. Bierce, A. Feder, G. et.al. (2020) Trauma-informed approaches to primary and community mental health care: protocol for a systematic review, BMJ Open 2020 2. Fallot, R. and Harris, M. (2009) In: Humanising Mental Health Care in Australia – A Guide to Trauma-Informed Approaches; Chapter 23, Pg. 307-318, Eds: Benjamin R, Haliburn JM, King, S. Routledge, 2019) 3. Jennings A (2004). Models for Developing Trauma-Informed Behavioural Health Systems and Trauma-Specific Services. Report. USA: National Association of State Mental Health Program Directors and the National Technical Assistance Centre for State Mental Health Planning.
--	--

REVIEWER	Robey Champine Michigan State University College of Human Medicine, Division of Public Health
REVIEW RETURNED	30-Nov-2020

GENERAL COMMENTS	The manuscript describes a protocol for a systematic review that will synthesize information about trauma-informed approaches to primary and community mental health care. Although the manuscript addressed an important topic, it contained substantive and stylistic issues that need to be addressed, as summarized below.  •Why does the timeline for the systematic review end in February 2020? It may be worthwhile to expand it to include December 2020. •According to the protocol, “there will be no language restrictions, provided an English language abstract is available for initial screening” (page 15 of PDF, lines 15-17). However, how will full-text articles that are not published in the English language be reviewed? This sentence seems to conflict with others in the manuscript suggesting that only studies published in the English language will be reviewed. •In your description of the “data extraction” approach, can you elaborate on the types of theories you will be coding for that underlie a TI approach (page 17 of PDF, line 29)? •Can you specify the steps involved in using the “results-based convergent synthesis design” that is mentioned (page 18 of PDF, lines 33-35)? •There were typos throughout the manuscript that were distracting. I stopped editing for them on page 8 of the PDF. •Define terms the first time they are introduced. For example, vicarious trauma (page 8 of PDF, line 59). •In the Abstract, change “leads” to “may lead to,” since there is variation in how individuals respond to potentially traumatic events (page 5 of PDF, line 11). •In the Abstract, change “including poorer mental and physical health, economic, social outcomes, and cognitive functioning to “including poor mental and physical health, economic, social, and cognitive functioning outcomes” (page 5 of PDF, lines 13-15). •Define acronyms the first time they are introduced (e.g., NHS on page 5 of PDF, line 27). •Include the word “and” before “databases” (page 5 of PDF, line 43). •Include “a” before “second reviewer” (page 5 of PDF, line 53). •Hyphenate “trauma informed” (page 7 of PDF, line 9). •Include “an” after the word “as” (page 7 of PDF, line 9).
--

	 •Change “In the World Health Organisation...” to “According to the World Health Organisation...” (page 7 of PDF, line 19). •Fix the typo in the following sentence: “...that occurred to a loved ones/witnessed...” (page 7 of PDF, line 25). •Insert a comma after “18%” (page 7 of PDF, line 35). •Change “is” to “was” (page 7 of PDF, line 41). •Replace “traumas” with “trauma” (page 7 of PDF, line 45). •Replace “lifespan” with “life span” (page 7 of PDF, line 51 and page 8 of PDF, line 43). The words lifespan and life-span are used to modify nouns. •Avoid the use of indefinite references. “This” what (page 7 of PDF, line 51)? •Insert “a” before “health system” (page 7 of PDF, line 55). •Insert “the” before “integration” (page 8 of PDF, line 11). •Insert “of” before “mental health” (page 8 of PDF, line 11). •Replace “improving” with “improved” (page 8 of PDF, line 15). •Replace “on” with “of” (page 8 of PDF, line 21). •Replace “traumas” with “trauma” (page 8 of PDF, line 21 and throughout the manuscript). •Insert “a” before “general” (page 8 of PDF, line 27). •Insert “The” before “majority” (page 8 of PDF, line 37). •Include a citation for the sentence that begins “However, they often have the opposite effect...” (page 8 of PDF, lines 45-55). •Include a citation for the sentence that begins “TI approaches can be defined as...” (page 9 of PDF, lines 11-17).
--	--

REVIEWER	Jacob K.Tebes Yale School of Medicine; USA
REVIEW RETURNED	08-Dec-2020

GENERAL COMMENTS	BMJ Open 2020-042112 – Trauma-informed approaches to primary and community mental health care: protocol for a systematic review. This manuscript describes a protocol for the completion of a systematic review of trauma-informed approaches to primary and community mental health care. The manuscript is well-written and clearly specifies components of the proposed protocol. Additional strengths of the manuscript and systematic review include: PROSPERO registration; excellent specification of guidelines, tools, and templates to extract and synthesize data; well-specified draft logic model; and robust patient and public involvement. Below I discuss several areas for improving the manuscript and the systematic review. (1) Page 9, line 5 (NOTE: Page numbers refer to the PDF PAGE.) - The authors should provide more of a rationale for linking primary care and community mental health care. They do this briefly on page 8, emphasizing WHO’s 2007 call for integrating mental health services into primary care. However, there are other reasons for linking these two services, including: both involve ambulatory health services; both have considerable overlap in the types of co-occurring challenges that individuals experience when seeking primary care or mental health care; and both primary care and community mental health care each often involve integrating an array of services across providers. For example, primary care providers often must integrate patient care across specialists, whereas community mental health care often involves provision of several types of services – mental health, social and vocational rehabilitation, housing – across community
--

	providers. Expanding the rationale for linking these two types of services may help the authors identify shared and distinct qualities between primary and community mental health care so as to suggest explanations for differences observed and provide directions for future research. (2) Page 12, line 51– The authors indicate that they will exclude “primary and community mental health care not using TI approaches.” This makes is reasonable but also reveals a limitation of their review. In the past, TI approaches were often embedded in usual services but not labelled as such, especially prior to the term “TI” entering the lexicon. For example, community mental health clubhouse models have long adopted, a participatory TI approach but did not frame their work this way until recently. Similarly, sexual assault survivor groups in the 1990s used TI approaches without explicitly labeling them as such until later. Although the authors may not be able to assess that a service uses TI approaches without mentioning this explicitly, they should note this as a limitation of their review. (3) Page 16, line 5 – The authors propose using the publication date of the Harris and Fallot (2001) book on designing trauma-informed services systems as the start date for their review. Although this book was influential, it post-dates other developments that led to the wide acceptance of trauma-informed approaches, including: establishment of the National Child Traumatic Stress Network in the U.S. in 2000; the early discussion of trauma-informed care principles in Australia and New Zealand prior to that; and the growth of sexual assault survivor networks as healing environments in the 1990s. The authors may consider beginning their review in 1990 even though it would not add many articles but would provide better coverage of their subject matter. (4) Page 18, line 15 – The authors propose tracking “characteristics of participants” in their review. This makes excellent sense, but they should specify what this entails. We know that trauma disproportionately impacts groups that are historically marginalized, disenfranchised, and discriminated against, such as women; blacks, indigenous peoples, and other people of color; individuals with disabilities; and individuals in under-represented religious groups. To the extent feasible, their review should track these categories or at the very least note that they are not sufficiently specified in the literature, thus encouraging future researchers to do so.
--	---

VERSION 1 – AUTHOR RESPONSE

Suggestion, question, or comment from reviewers	Author’s response
Reviewer 1- Joan Marie Haliburn	
1. A systematic review of synthesized evidence in trauma-informed approaches to health care in primary and community mental health facilities globally, is necessary, in fact	Thank you for acknowledging the importance and timeliness of our systematic review.

Suggestion, question, or comment from reviewers	Author's response
imperative, if we are to truly deliver timely assessment and treatment without re-traumatizing individuals who come for help, or forgetting workers who may run the risk of vicarious traumatization. This review is indeed timely and can be an innovative contribution to the mental health care literature and to mental health care in general. Trauma-informed care is an emergent paradigm, but the current organization of health and human services does not reflect this reality and is inadequate, thus a common finding is one of trauma occurring in a service itself (Jennings 2004). Results of a well-thought through study will help to make sure that services carry a detailed definition of what it is to be trauma-informed and what it is to deliver a service that adheres to this definition. The impact of trauma was recognized decades ago, but the roots of trauma-informed care are more recent, and we must be sure that it delivers this care based on well-informed principles. Trauma-informed care is being incorporated in policies, world-wide, but where is the evidence? This review seeks to provide us with the evidence. This study has sought input from those with lived experience of trauma, and professionals who work with those who have been traumatized. In doing so it has respected the knowledge that sufferers themselves have experienced – both trauma as well as care – thus being able to attest to the kind of care they have received, and whether they felt it was adequate or not – the flaws along with the positive aspects and suggestions for improvement. It also intends to seek the personal experiences of professionals who have delivered services. Professionals similarly can be gratified to know that their interests are also in the minds of employment systems.	
2. However, this study seeks to adopt the definition of trauma-informed care used by SAMHSA the leading institution in this field. In this definition, the words psychological, emotional and behavioural are missing (Dawson et. al 2020: p. 5, lines 16- 22), though this is made good in the Introduction (Dawson et.al 2020: p. 7 lines 9-18). They are the core issues in both assessment and treatment of patients in the mental health system. SAMHSA since 1984 has pursued the need for trauma-informed services and has since become a major vehicle for the	Our preliminary work found varied definitions of TIC. We have quoted the consensus-based definition of TI care by Hopper et al (2010) which includes psychical, psychological and emotional safety of patients and providers (Introduction, para 2, pg. 7). We acknowledge that the SAMHSA framework for trauma and TI care was proposed for the behaviour health speciality sectors. However, the authors highlighted that their framework is

Suggestion, question, or comment from reviewers	Author's response
advocacy of Trauma Informed Care in the USA. Trauma “has often occurred in the service context itself” (Jennings, 2004: p.6). Will the principles of trauma-informed care provided by drug and alcohol services fit with the principles of those facilities providing care of the rest of the mentally ill?	not a prescribed set of practices and procedures but rather a set of fundamental assumptions and principles that can be adapted to varied settings and sectors including primary health care (SAMHSA's Concept of Trauma and Guidance for a Trauma-Informed Approach 2018, p. 3, para 2) We found several adaptations of the SAMHSA framework to varied settings, all referencing the SAMHSA' foundational assumptions, principles, and implementation domains. The terminology may differ across sectors and settings, but the core components align with the SAMHSA framework. We added this information to Introduction (Introduction, para 2, pg. 8)
3. The choice of primary health care and community health care as sources of study is a good one, as increased health care utilization is strongly connected with early childhood traumas. Therefore, the need to take particular precautions in assessment and treatment are vital, and so is the need to prevent vicarious re-traumatization.	Thank you for acknowledging the importance of undertaking this study in primary care and community mental health care.
4. Trauma informed approaches to care shift the health care professional's focus from diagnosis towards understanding trauma that lies at the root of symptoms and health conditions (page 9, lines 17-43). Awareness of a trauma history, or awareness of trauma in a person's mental state examination needs to be borne in mind and addressed alongside the presenting condition - but being seen as “the root of symptoms and health conditions” deserves a more cautious approach. We still do not have the evidence that trauma is at the root of Schizophrenia and other Major Mental Disorders, even though the associations are high.	We agree that the evidence base for trauma being a root of major mental disorders is in development and softened relevant statements throughout the manuscript saying that trauma may lead to a range of effects on survivors.
5. Principles of Trauma-Informed Care must take priority in any trauma-informed service, and adherence to such principles can become apparent in the manner in which organizations teach, train and observe such principles, both towards their staff as well as in the delivery of services to their patients. Observations of principles such as safety, trust, choice, collaboration and empowerment are necessary if a service is to be truly trauma -Informed. This could do with some elaboration in the study.	We have added the SAMHSA's six key principles of a TI care (safety, trustworthiness and transparency, peer support, collaboration and mutuality, empowerment, and cultural/historic/gender issues) to Introduction (Introduction, para 1, pg. 8).

Suggestion, question, or comment from reviewers	Author's response
6. This review on the whole takes into consideration the essentials required in trauma informed services which are widely advocated but require sound evidence in its implementation.	We agree. Our review addresses the gap between policies endorsing implementation of TIC and limited evidence on the effectiveness and cost-effectiveness of TIC.
7. This study promises to be an innovative contribution to the mental health literature. Trauma-informed-care are the catch words in current mental health services, but there is no certainty that they are truly trauma informed i.e., adhering to the principles of such an approach. It will be a step towards refining such principles, leading to a more global understanding and approach. The study proposes to use the definition of trauma as proposed by SAMHSA, the leading institution in the field of trauma-informed-care. Do you think their definition will suit other mental health services entirely? The authors have wisely chosen to obtain data from primary mental health services - they are usually the first point of contact, and the amount of data gathered will be significant to yield valid results, and the diversity of practitioner will enrich it.	We think that the SAMHSA definition of trauma is adaptable to mental health services delivered in primary and community mental health settings (see response to comment 2 above).
Reviewer 2- Robey Champine	
8. Why does the timeline for the systematic review end in February 2020? It may be worthwhile to expand it to include December 2020.	The timeline for the systematic review ends in February 2020 as this was the timepoint at which we submitted our protocol for registration with PROSPERO.
9. According to the protocol, "there will be no language restrictions, provided an English language abstract is available for initial screening" (page 15 of PDF, lines 15-17). However, how will full-text articles that are not published in the English language be reviewed? This sentence seems to conflict with others in the manuscript suggesting that only studies published in the English language will be reviewed.	Thank you for pointing this out. We will be inclusive at the title and abstract screening stage and include studies at this stage if they meet the inclusion criteria. During full-text screening stage, if the full-text is not available in English, we will translate it with help from multilingual colleagues and Google Translate. (see Language under eligibility criteria, pg. 12).
10. In your description of the "data extraction" approach, can you elaborate on the types of theories you will be coding for that underlie a TI approach (page 17 of PDF, line 29)?	This systematic review will inform development of a theory of change for a UK specific TIC approach (Introduction, para 2, pg. 8). We want to evaluate if the included interventions have been supported by formal theories and/or empirically developed theories of change explaining why and how they work. We are

Suggestion, question, or comment from reviewers	Author's response
	also interested in whether the included evaluations were theory-informed to understand why interventions produced or did not produce expected outcomes. We have clarified this in the second research question (para 1, pg. 9), outcomes (last para pg. 10 and draft logic model supplementary file). If authors reported any theories underpinning their interventions and/or evaluations, we will extract data on which theories were used and how. If this information is not reported, we will ask corresponding authors for this information. If authors reported evidence of a mechanism, enabler or barrier to improving any outcomes of interest for this systematic review, we will extract this data to inform our theory of change. We have added this explanation to the data extraction section (Data extraction, pg. 14).
11. Can you specify the steps involved in using the “results-based convergent synthesis design” that is mentioned (page 18 of PDF, lines 33-35)?	We appreciate the Reviewer pointing out the need to specify the steps involved in using this approach. We have rewritten this section to offer more clarity, we have provided additional information (see Data synthesis section, pg. 15 and para 1, 16).
12. •There were typos throughout the manuscript that were distracting. I stopped editing for them on page 8 of the PDF.  •Define terms the first time they are introduced. For example, vicarious trauma (page 8 of PDF, line 59). •In the Abstract, change “leads” to “may lead to,” since there is variation in how individuals respond to potentially traumatic events (page 5 of PDF, line 11). •In the Abstract, change “including poorer mental and physical health, economic, social outcomes, and cognitive functioning to “including poor mental and physical health, economic, social, and cognitive functioning outcomes” (page 5 of PDF, lines 13-15). •Define acronyms the first time they are introduced (e.g., NHS on page 5 of PDF, line 27). 	We would like to thank the reviewer for pointing this out. We have now made all the suggested changes and have proofread the document to ensure that there are no further typos. Thank you for this suggestion. As TIA is the central focus of this review, we have only defined what we mean by this as this and have decided to not define every term discussed in the introduction. However, we have provided references to seminal texts that define vicarious trauma and other common phenomenon.

Suggestion, question, or comment from reviewers	Author's response
 •Include the word “and” before “databases” (page 5 of PDF, line 43). •Include “a” before “second reviewer” (page 5 of PDF, line 53). •Hyphenate “trauma informed” (page 7 of PDF, line 9). •Include “an” after the word “as” (page 7 of PDF, line 9). •Change “In the World Health Organisation...” to “According to the World Health Organisation...” (page 7 of PDF, line 19). •Fix the typo in the following sentence: “...that occurred to a loved ones/witnessed...” (page 7 of PDF, line 25). •Insert a comma after “18%” (page 7 of PDF, line 35). •Change “is” to “was” (page 7 of PDF, line 41). •Replace “traumas” with “trauma” (page 7 of PDF, line 45). •Replace “lifespan” with “life span” (page 7 of PDF, line 51 and page 8 of PDF, line 43). The words lifespan and life-span are used to modify nouns. •Avoid the use of indefinite references. “This” what (page 7 of PDF, line 51)? •Insert “a” before “health system” (page 7 of PDF, line 55). •Insert “the” before “integration” (page 8 of PDF, line 11). •Insert “of” before “mental health” (page 8 of PDF, line 11). •Replace “improving” with “improved” (page 8 of PDF, line 15). •Replace “on” with “of” (page 8 of PDF, line 21). •Replace “traumas” with “trauma” (page 8 of PDF, line 21 and throughout the manuscript). •Insert “a” before “general” (page 8 of PDF, line 27). •Insert “The” before “majority” (page 8 of PDF, line 37). •Include a citation for the sentence that begins “However, they often have the opposite effect...” (page 8 of PDF, lines 45-55). •Include a citation for the sentence that 	

Suggestion, question, or comment from reviewers	Author's response
begins "TI approaches can be defined as..." (page 9 of PDF, lines 11-17).	We included references for these two statements.
Reviewer 3- Jacob K. Tebes	
13. This manuscript describes a protocol for the completion of a systematic review of trauma-informed approaches to primary and community mental health care. The manuscript is well-written and clearly specifies components of the proposed protocol. Additional strengths of the manuscript and systematic review include: PROSPERO registration; excellent specification of guidelines, tools, and templates to extract and synthesize data; well-specified draft logic model; and robust patient and public involvement. Below I discuss several areas for improving the manuscript and the systematic review. (1) Page 9, line 5 (NOTE: Page numbers refer to the PDF PAGE.) - The authors should provide more of a rationale for linking primary care and community mental health care. They do this briefly on page 8, emphasizing WHO's 2007 call for integrating mental health services into primary care. However, there are other reasons for linking these two services, including: both mostly involve ambulatory health services; both have considerable overlap in the types of co-occurring challenges that individuals experience when seeking primary care or mental health care; and both primary care and community mental health care each often involve integrating an array of services across providers. For example, primary care providers often must integrate patient care across specialists, whereas community mental health care often involves provision of several types of services – mental health, social and vocational rehabilitation, housing – across community providers. Expanding the rationale for linking these two types of services may help the authors identify shared and distinct qualities between primary and community mental health care so as to suggest explanations for differences observed and provide directions for future research.	We have added other reasons for focusing on the primary healthcare and community mental health care (Introduction, para 1, pg. 6)
14. (2) Page 12, line 51– The authors indicate that they will exclude "primary and community mental health care not using TI approaches."	Our search terms are based on the current TI lexicon and include terms 'trauma*', 'adverse

Suggestion, question, or comment from reviewers	Author's response
This makes is reasonable but also reveals a limitation of their review. In the past, TI approaches were often embedded in usual services but not labelled as such, especially prior to the term "TI" entering the lexicon. For example, community mental health clubhouse models have long adopted, a participatory TI approach but did not frame their work this way until recently. Similarly, sexual assault survivor groups in the 1990s used TI approaches without explicitly labeling them as such until later. Although the authors may not be able to assess that a service uses TI approaches without mentioning this explicitly, they should note this as a limitation of their review.	childhood**' and 'psychologically informed environment**' (Appendix 2, supplementary file). The last term was proposed by our professional advisory group as a pre-TI label. We have acknowledged this terminology limitation (Discussion, para 2, pg. 17)
15. (3) Page 16, line 5 – The authors propose using the publication date of the Harris and Fallot (2001) book on designing trauma-informed services systems as the start date for their review. Although this book was influential, it post-dates other developments that led to the wide acceptance of trauma-informed approaches, including: establishment of the National Child Traumatic Stress Network in the U.S. in 2000; the early discussion of trauma-informed care principles in Australia and New Zealand prior to that; and the growth of sexual assault survivor networks as healing environments in the 1990s. The authors may consider beginning their review in 1990 even though it would not add many articles but would provide better coverage of their subject matter.	Thank you for this suggestion. We have now extended the search period to cover January 1990 (Time frame under eligibility criteria pg. 12)
16. (4) Page 18, line 15 – The authors propose tracking "characteristics of participants" in their review. This makes excellent sense, but they should specify what this entails. We know that trauma disproportionately impacts groups that are historically marginalized, disenfranchised, and discriminated against, such as women; blacks, indigenous peoples, and other people of color; individuals with disabilities; and individuals in under-represented religious groups. To the extent feasible, their review should track these categories or at the very least note that they are not sufficiently specified in the literature, thus encouraging future researchers to do so.	Thank you for this suggestion. We have now provided further information to elaborate on participant characteristics (Data extraction, pg. 14)

VERSION 2 – REVIEW

REVIEWER	JOAN HALIBURN. CONSULTANT PSYCHIATRIST LECTURER, UNIVERSITY OF SYDNEY
REVIEW RETURNED	26-Jan-2021

GENERAL COMMENTS	Congratulations on a well thought through paper. Yes, trauma-informed approaches are advocated, but what does it mean to be trauma-informed and is a service truly trauma informed depends upon a number of different parameters. This attempt to study and have some definite parameters established by services is important and overdue. The evidence base for trauma-informed approaches to health care is not well established. A systematic review of synthesised evidence in trauma-informed approaches in primary care and community mental health care globally is necessary to properly inform development of such services. This paper is a robust attempt to systematically study the services that purport to deliver trauma-informed services – in order to see if they do what they say they do. Using those adults who have lived-experience as well as professionals who plan and deliver these trauma-informed services in developing the study is also a good idea. Trauma-informed care, trauma-informed services and trauma-informed attitudes are the emergent paradigms. We have seen through the ages, organisations and individuals pay lip-service to what is emergent, and unfortunately what is trendy. It behoves us to make sure that the delivery of such services adheres to the principles elucidated. This study aims to do just that. It is a promising study, and an innovative contribution to the health-care literature. Current organisations of health and human services do not reflect this reality and are quite inadequate. Trauma has often occurred in the service delivery itself, the impact of trauma on service deliverers has long been known, but little attended to. A cultural shift is required from the top, down, and be accountable. A study such as this has the capacity to be replicated and provide impetus for those who feel similarly about the need for real trauma-informed care
---

REVIEWER	Jacob Tebes Yale School of Medicine, USA
REVIEW RETURNED	31-Jan-2021

GENERAL COMMENTS	The authors have satisfactorily addressed the concerns previously raised in the reviews.
--